# TDP-43 and Alzheimer’s Disease Pathology in the Brain of a Harbor Porpoise Exposed to the Cyanobacterial Toxin BMAA

**DOI:** 10.3390/toxins16010042

**Published:** 2024-01-12

**Authors:** Susanna P. Garamszegi, Daniel J. Brzostowicki, Thomas M. Coyne, Regina T. Vontell, David A. Davis

**Affiliations:** 1Department of Neurology, Miller School of Medicine, University of Miami, Miami, FL 33136, USA; 2Department of Pathology, Immunology and Laboratory Medicine, University of Florida, Gainesville, FL 32610, USA

**Keywords:** blue green algae bloom, cetacean stranding, Guam ALS/PDC, marine neurotoxin, marine food web, toothed whales

## Abstract

Cetaceans are well-regarded as sentinels for toxin exposure. Emerging studies suggest that cetaceans can also develop neuropathological changes associated with neurodegenerative disease. The occurrence of neuropathology makes cetaceans an ideal species for examining the impact of marine toxins on the brain across the lifespan. Here, we describe TAR DNA-binding protein 43 (TDP-43) proteinopathy and Alzheimer’s disease (AD) neuropathological changes in a beached harbor porpoise (*Phocoena phocoena*) that was exposed to a toxin produced by cyanobacteria called β-N-methylamino-L-alanine (BMAA). We found pathogenic TDP-43 cytoplasmic inclusions in neurons throughout the cerebral cortex, midbrain and brainstem. P62/sequestosome-1, responsible for the autophagy of misfolded proteins, was observed in the amygdala, hippocampus and frontal cortex. Genes implicated in AD and TDP-43 neuropathology such as *APP* and *TARDBP* were expressed in the brain. AD neuropathological changes such as amyloid-β plaques, neurofibrillary tangles, granulovacuolar degeneration and Hirano bodies were present in the hippocampus. These findings further support the development of progressive neurodegenerative disease in cetaceans and a potential causative link to cyanobacterial toxins. Climate change, nutrient pollution and industrial waste are increasing the frequency of harmful cyanobacterial blooms. Cyanotoxins like BMAA that are associated with neurodegenerative disease pose an increasing public health risk.

## 1. Introduction

Cetaceans are considered one of the most intelligent species on earth with self-awareness, long-term memory and social recognition that can last for at least two decades [1]. They are long-lived mammals, with some species having an encephalization quotient that is far closer to humans than non-human primates and the great apes [2]. The complexity of their neuroanatomy, neurocircuitry, acoustic communication, sleep and social behavior has been a focus of research for many decades [3,4,5]. Recently, emerging evidence suggests that several cetacean species can also develop age-related neurodegenerative pathological changes reminiscent of those observed in Alzheimer’s disease (AD) [6,7,8,9,10,11]. Cetaceans are also a well-regarded sentinel species for toxin exposures forewarning of the accumulation of toxic compounds in the marine environment [12,13,14]. Since cetaceans can develop AD-like neuropathological changes and are exposed to marine toxins throughout their life span, they are an ideal model species to assess the long-term effects of neurotoxins on brain health. 

β-N-methylamino-L-alanine (BMAA) is a non-canonical amino acid made by 95% of cyanobacteria genera [15,16]. Chronic dietary exposure to BMAA is associated with neurodegenerative disease in humans [17]. In the western Pacific, Guamanians with diets high in BMAA developed a complex neurodegenerative disease call *Lytico-bodig* or amyotrophic lateral sclerosis/parkinsonism-dementia complex (ALS/PDC), which consists of motor neuron degeneration and dementia [18,19,20]. The neuropathological changes in ALS/PDC include abundant cortical neurofibrillary tangles (NFTs), sparse-to-moderate amyloid-β (Aβ) plaques and TAR DNA-binding protein 43 (TDP-43) cytoplasmic inclusions (CI) [21,22]. These autopsy findings were further supported by non-human primate models that were fed high doses of BMAA and developed TDP-43 CI, NFTs and Aβ deposits in their brain and spinal cord [23,24,25]. Outside of Guam, the BMAA toxin has been detected throughout the marine food web, including the brain and muscle tissues of apex predators [26,27,28]. BMAA has also been detected in autopsied brain samples and CSF of individuals with AD and ALS [29,30,31]. Thus, the presence of BMAA throughout the marine food web is a concern for apex predators and humans at risk of developing dementia and motor neuron disease. Because the detection of BMAA in human tissue can vary due to a number of factors, examining marine mammals that have a lifetime risk of cyanotoxin exposure may provide more consistent measurements and allow for novel insights into BMAA’s mechanism of neurotoxicity [20,29,30,32,33]. 

We have previously reported that the BMAA toxin can bioaccumulate in the cerebral cortex of two dolphin species: *Tursiops truncatus* and *Delphinus delphis* [7]. The same dolphins displayed neuropathological changes reminiscent of AD and the severity of these neuropathological changes increased with BMAA exposure [6]. These studies suggest BMAA may have a role in the development of neurodegenerative phenotype in cetaceans. Here, we present a neurotoxicological analysis of a postmortem brain procured from a harbor porpoise (*Phocoena phocoena*) found beached in Cape Cod Bay, MA, USA. BMAA testing, qPCR analysis for genes implicated in dementia and motor neuron diseases and neuropathological evaluation were performed on corresponding brain regions implicated in AD and TDP-43 proteinopathies.

## 2. Results

### 2.1. Case History

In March 2012, a live female nonpregnant subadult harbor porpoise was reported stranded alone at Chapin Beach in Dennis, MA, USA (condition code 1) (LAT: 41.72697; LONG: −70.23891). The harbor porpoise was found in critical condition displaying signs of emaciation and scavenger trauma to the left eye caused by sea gulls. No human interactions were observed based on physical assessment. The harbor porpoise (ID: IFAW 12-206 Pp) was determined to be in poor health and was euthanized, followed by an necropsy where the weight (19 kg) and length were measured (1.1 m). The brain was removed, preserved and provided to the University of Miami Brain Endowment Bank (UMBEB) for gross, neuropathological and toxicological analysis. 

### 2.2. Cyanobacterial Toxin BMAA Exposure

We detected BMAA in two adjacent cortical gyri of the parietal cortex (Pc) of IFAW 12-206 Pp: the auditory cortex or A1 (Pc^A1^) (93.6 + 9.1 μg/g) and the visual cortex of V1 (Pc^V1^) (93.1 + 7.5 μg/g). BMAA tissue concentrations did not differ between the two cortical gyri (Table 1). We also detected BMAA structural isomers: 2,4-diaminobutyric acid (2,4-DAB) and N-(2-aminoethyl)glycine (AEG). However, AEG was not detected in Pc^V1^ (*p* = 0.001, *t*-test) (Table 1). BMAA was 5.7 times more concentrated than AEG and 2.8 times less concentrated than 2,4-DAB. In addition, we observed a positive correlation between 2,4-DAB and BMAA brain tissue concentrations (r = 0.7964; *p* = 0.01, Pearson r). No correlation with BMAA and AEG concentration was observed (r = −0.2278; *p* = 0.4692, Pearson r). 

### 2.3. Expression of Genes Implicated in Alzheimer’s Disease and TDP-43 Proteinopathy

All dementia-related genes were expressed in each brain region analyzed from IFAW 12–206 Pp (Table 2). Six of the seven genes evaluated were expressed at higher levels in the cerebral cortex (Crtx) when compared to the cerebellum (CE), a region mostly unaffected in AD, ALS and frontal temporal dementia (FTD). One gene, presenilin-2 (*PSEN2*), was decreased in both Crtx regions. Gene expression patterns for Pc^V1^ and frontal cortex (Fc) were positively correlated (r = 0.844; *p* = 0.017, Pearson r) and mirrored the region-specific expression of genes in the Pc and prefrontal cortex (PreFc) of humans. The Aβ precursor protein (APP) (2.7-fold; *p* < 0.0001, ANOVA) and microtubule-associated protein tau (*MAPT*) (2.4-fold; *p* = 0.0134, ANOVA) were the most highly expressed genes in the Crtx. *TARDBP*, the gene that encodes for TDP-43 protein, was also expressed up to ~2.0-fold (*p* = 0.0551, ANOVA) more in the Crtx. These data suggest that the harbor porpoise displayed similar brain region-specific gene expression patterns as in humans. 

**Table 2 toxins-16-00042-t002:** Comparative brain region specific gene expression of dementia related genes in the harbor porpoise.

Gene Probes	Neurological Diseases	Associated Neuropathology	Harbor Porpoise (qPCR)		Human (BioGPS)	
			Pc^V1^:CE	Fc:CE	Pc:CE	PreFc:CE
*APP* [34]	AD	Aβ^+^ plaques	1.41 ± 0.34	2.88 ± 0.56 *^p^* ^< 0.0001^	1.73	4.20
*PSEN1* [35]	AD	Aβ^+^ plaques	1.23 ± 0.02	1.33 ± 0.03	7.48	7.56
*PSEN2* [36]	AD	Aβ^+^ plaques	0.66 ± 0.03	0.79 ± 0.01	1.30	1.46
*GRN* [37]	FTLD	progranulin loss	1.38 ± 0.02	1.96 ± 0.04	1.24	1.42
*MAPT* [38]	Tauopathies	NFTs	1.52 ± 0.07	2.40 ± 0.14 *^p^* ^= 0.0134^	4.08	7.24
*TARBDP* [39]	TDP-43 proteinopathies	IC	1.21 ± 0.01	1.95 ± 0.04	1.40	2.28
*C9orf72* [39]	ALS and FTD	IC	1.22 ± 0.09	1.66 ± 0.09	1.03	0.96

Aβ: amyloid beta; AD: Alzheimer’s disease; ALS: amyotrophic lateral sclerosis; CE: cerebellum (baseline); Fc: frontal cortex; FTD: frontotemporal dementia; FTLD: frontotemporal lobar degeneration; IC: intracytoplasmic inclusions; Pc^V1^: visual cortex; PreFc: prefrontal cortex.

### 2.4. Neuropathology 

Gross examination of the left-brain hemisphere demonstrated leptomeningeal congestion and cerebral edema characterized by cortical gyral widening and sulcal effacement. The Crtx, CE, midbrain (Mid) and brainstem (Bs) were intact. The external examination revealed a normal gyral configuration without evidence of contusions or hemorrhage, cystic degeneration or xanthochromic discoloration (Figure 1A,B). The cerebrum was sectioned in the coronal plane. The cortical ribbon was of normal thickness and displayed appropriate gray-white matter demarcation without gross dysplasia, heterotopia or laminar necrosis. The subcortical white matter was congested, and the deep nuclei were symmetrical without infarction, hemorrhage or mass lesions. The cerebellum and brainstem had a normal gross architecture without infarction or lesions (Figure 1C). For comparison, examples of normal cetacean gross neuroanatomy can be seen in brain atlases (see the Methods section). Diffuse-type phospho-TDP-43 CI were observed throughout the Crtx, Mid and Bs (Figure 2). TDP-43 CI were predominantly found in the perikaryon of neurons in the Crtx cortical layers II-VI. The occipital cortex (Oc) was negative for TDP-43 CI (Figure 2F, Table 3). In the hippocampus (Hipp), AD-type changes such as Hirano bodies [40], granulovacuolar degeneration bodies (GVBs) [41], intracellular tangles, ghost tangles and dense-core plaques [42] were present (Figure 3). Microtubule-associated protein 2 (MAP2) immunostaining highlighted degenerated neurons in the Cornu Ammonis areas of the Hipp and the Fc (Figure 3M,N). Aβ^+^ plaques and intraneuronal staining was observed in all brain regions analyzed in this study (Figure 3O, Table 3). Moderate numbers of the Aβ^+^ diffused and dense core plaques were wide-spread and varied in size (Figure 3P–R). Sequestosome 1 (P62/SQSTM1) staining was found in brain regions involved in the progression and neuropathological staging of AD and the CE (Table 3). P62/SQSTM1 staining was not observed in the Med, Pons, temporal cortex (Tc), Oc or Pc (Table 3). Hippocampal CA2 neurons, responsible for the formation of social memory, had numerous argyrophilic-, TDP-43^+^, Aβ^+^, P62/SQSTM1^+^ intraneuronal inclusions and Aβ^+^ plaque deposition (Figure 3S-V). These data suggest that the harbor porpoise can develop TDP-43 proteinopathy with co-current AD neuropathological changes. 

## 3. Discussion

Climate change, nutrient pollution and industrial waste are increasing the numbers of harmful cyanobacterial blooms (HCBs) [43,44]. HCBs produce cyanotoxins which can have wide-ranging effects on marine life, public health and the economy [45]. The impact of HCBs can vary depending on the region [46]. However, the harmful effects of some cyanotoxins, termed “slow toxins”, can manifest neurological symptoms in an individual decade after exposure [18]. Here, we examined the potential impact of cyanotoxin exposure on a harbor porpoise, a species under increased metabolic stress due to their size, making them especially vulnerable to toxicity and mortality [47,48]. We detected BMAA, a neurotoxin produced by cyanobacteria, in the brain at concentrations comparable to individuals with AD, ALS and ALS/PDC [20,29]. We also detected two BMAA structural isomers: 2,4-DAB and AEG. We examined samples taken from two adjacent cortical gyri in the parietal cortex and did not observe a difference in BMAA or 2,4-DAB concentrations, but we did observe a difference in AEG. In vitro models have shown that all three compounds have different mechanisms of neurotoxicity [49]. Thus, the combined neurotoxic effects of all three compounds may be under estimated. More studies are needed to understand the regional distribution, differential concentrations and additive or synergistic effects of BMAA and BMAA isomers [50,51]. 

We also show here that the harbor porpoise express genes that are involved in the pathogenesis of AD and TDP-43 proteinopathies. Expression levels were highest in the frontal cortex and consisted of genes involved in the deposition of Aβ^+^ plaques, NFTs and TDP-43 CI. The amyloid-β precursor protein (*APP*) was the most highly expressed gene and was accompanied by Aβ^+^ deposits and plaques throughout the brain. In our analysis, we did not evaluate if mutations were present in the genes analyzed. However, the similarities in the differential expression of genes in the cortex versus the cerebellum suggest these genes may contribute to the development of neuropathology in the harbor porpoise as they do in humans. These findings along with previous studies from our group support the need for further research into understanding the impact of gene and cyanotoxin interactions on the cetacean brain [6]. 

BMAA exposure is a strong risk factor in the development of TDP-43 proteinopathy [52]. In non-human primates and rodents, BMAA exposure increases the deposition of TDP-43 cytoplasmic inclusions [23,25]. The co-occurrence of TDP-43 proteinopathy has been shown to increase the onset, severity and progression of AD [53]. Pathological TDP-43 causes greater atrophy of the hippocampus in AD, a brain region responsible for learning and memory [54]. The presence of TDP-43 CI is also a pathological hallmark of diseases such as ALS, FTD and limbic-predominant age-related TDP-43 encephalopathy (LATE) [55,56]. Here, we observed TDP-43 CI throughout the brain of a harbor porpoise exposed to BMAA. TDP-43 CI were found in the cerebral cortex, hippocampus and brainstem similar to the distribution reported in ALS/PDC [57]. Additional studies are necessary to understand the link between BMAA exposure, TDP-43 proteinopathies and if lesions are primary or secondary hallmarks of a neurodegenerative phenotype in the harbor porpoise.

We observed intraneuronal staining of P62/sequestosome 1 (P62/SQSTM1) a marker of autophagy of misfolded proteins. In AD, P62/SQSTM1 is associated with the early phases of NFT development [58]. Mutations in P62/SQSTM1 have been associated with ALS [59]. P62/SQSTM1 immunoreactivity was observed in neurons of brain regions implicated in the clinical progression of AD. The presence of TDP-43 CI, P62/SQSTM1, Aβ and NFTs in CA2 hippocampal neurons, a region responsible for social memory, suggest neurotoxicity in the development of a progressive neurodegenerative disease [60,61]. Further studies are needed to examine other protein markers related to neurodegenerative disease in order to better understand the heterogeneity of pathology in the harbor porpoise. 

Here, we also show two neuropathological hallmarks of AD: Aβ^+^ plaques and NFTs. Aβ^+^ plaques were observed throughout the brain regions analyzed and its immunoreactivity was robust using the β-Amyloid clone 6E10 antibody. NFTs were also widespread, consisting of pre-, mature and ghost or tombstone tangles similar to those observed in the common dolphin [6]. However, NFTs were identified in the harbor porpoise brain using Sevier Münger silver staining due to the negative immunoreactivity to AT8 (phospho-tau at serine 202, threonine 205) and the AT180 (phospho-tau at threonine 231) antibody that was reported by Vacher et al. [10]. These finding suggest other tau phospho-sites may be involved in the disease process in the harbor porpoise. Additional research is needed to delineate the phosphorylated tau sites involved in NFT formation in the brain of the harbor porpoise. 

HCBs produce a number of cyanotoxins that affect the nervous system [62]. Experimental models have shown that the co-exposures to cyanobacterial toxins can be synergistic [63,64,65,66,67]. Compounds concentrated in the marine food web like methylmercury that has been shown to have synergistic effects with BMAA should also be considered when evaluating neurotoxicity in cetaceans [6,51,68,69]. Furthermore, infectious disease agents such as herpesvirus, morbillivirus, *Brucella ceti* and prions that can contribute to neurodegeneration in cetaceans should be consider as a co-morbidity with cyanotoxin exposures [8,11,46,70,71,72,73]. Future studies will be needed to assess the synergistic effects of cyanotoxin exposures and co-current neurological diseases on triggering TDP-43 and AD pathology.

## 4. Conclusions

The cyanotoxins BMAA, AEG and 2,4-DAB, were detected in the brain of a beached harbor porpoise. The harbor porpoise also displayed severe TDP-43 proteinopathy with AD-type neuropathological changes. These findings further support the development of a neurodegenerative phenotype in cetaceans and a potential causative link to chronic cyanotoxin exposure.

## 5. Materials and Methods

### 5.1. Harbor Porpoise Brain

A female subadult harbor porpoise (*Phocoena phocoena*) found beached in Massachusetts in March of 2012 was examined in this study. Physical assessments performed on-site by the stranding response team determined the harbor porpoise was in poor health and should be subsequently euthanized. The harbor porpoise (ID: IFAW 12-206 Pp) was not euthanized for this research study. A necropsy was performed the next day at the Woods Hole Oceanographic Institute Marine Research Facility (WHOI MRF) and the International Fund for Animal Welfare (IFAW). The age class of IFAW 12-206 Pp was estimated as described in Geraci et al. [74]. For neurotoxicological analysis, the brain was removed and the right hemisphere was frozen. The contralateral brain hemisphere was fixed in 10% buffered formalin. After preservation, the whole brain was shipped to the University of Miami Brain Endowment Bank (UMBEB) for further analysis. Upon receipt, frozen parietal cortex-visual cortex area 1 (Pc^V1^), frontal cortex (Fc) and the cerebellar hemisphere (CE) were dissected for qPCR analysis. Twelve brain regions were sampled from the left hemisphere for neuropathological evaluation: amygdala (Amy); *CE*; cingulate gyrus (Cg); entorhinal cortex (Ent); hippocampus (Hipp); medulla oblongata (*Med*); occipital cortex (Oc); parietal cortex (*Pc^A1^*), *Pc^V1^*, pontine (*Pons*); temporal cortex (Tc). Brain atlases by Breathnach and Goldby [75] and Marino et al. [76] were used for the identification of harbor porpoise neuroanatomy. The Michigan State University Brain Biodiversity Bank Dolphin Atlas was also used as another reference for normal cetacean neuroanatomy (https://brains.anatomy.msu.edu/brains/dolphin/index.html) accessed on 29 December 2023. Experiments in this manuscript were approved by the National Oceanic Atmospheric Administration (NOAA) Southeast Region Stranding Program and NOAA Fisheries Service. The University of Miami Institutional Animal Care and Use Committee (IACUC) reviewed and authorized this study before receiving brain specimens. The handling of the harbor porpoise tissues satisfies the requirements of the Marine Mammal Protection Act pursuant to 50 CFR 216.22.

### 5.2. HPLC/FD

High-performance liquid chromatography with fluorescence detection (HPLC-FD) as previously described in Davis et al. 2019 was used to measure BMAA in 150 mg of porpoise cortical brain tissue [7,29]. Briefly, BMAA was separated from two structural isomers 2,4-diaminobutyric acid (2,4-DAB) and (N-(2-aminoethyl)glycine (AEG) using reverse-phase elution on a 1525 Binary HPLC pump and a 717 autosampler (Waters Corp., Milford, MA, USA). Analytes of interests were separated at 33.0 min (2,4-DAB), 31.1 min (BMAA) and 29.6 min for (AEG). The detection of analytes utilized a 2475 Multi k-Fluorescence Detector (Waters Corp., USA) with excitation/emission at 250/395 nm, respectively. Replicate measurements (*n* = 4–5) were performed for each brain region (*Pc^A1^* and *Pc^V1^*) and compared to control samples containing known amounts of an L-BMAA standard (Sigma-Aldrich, St. Louis, MO, USA). The limit of detection and the limit of quantification for our analysis were 2.7 ng/mL and 7.0 ng/mL, respectively.

### 5.3. qPCR Analysis

Brain tissues samples from *Pc^V1^*, *Fc* and *CE* (100 mg) were dissected to extract the total RNA using an RNeasy Lipid Tissue Mini Kit and DNase I on-column treatment (Qiagen Inc., Germantown, MD, USA). RNA concentration (μg/μL) and quality (RIN) were measured for each sample using a NanoDrop 2000 Spectrophotometer (Thermo Fisher Scientific, Waltham, MA, USA) and an Agilent 2100 Bioanalyzer (Agilent Technologies Inc., Santa Clara, CA, USA). In this study, high-quality RNA samples were used for IFAW 12-206 Pp, which averaged a RIN of 9.7 out of 10. Total RNA (5 μg) and a High Capacity Reverse Transcription Kit (Thermo Fisher Scientific, Waltham, MA, USA) was used to create complementary DNA (cDNA) libraries for each brain region of interest. A custom dolphin PCR assay containing seven genes (*APP*, *PSEN1*, *PSEN2*, *MAPT*, *GRN*, *TARDBP* and *C9orf72*) involved in the development of AD and TDP-43 proteinopathies were created to measure gene expression using a TaqMan Universal PCR Master Mix on a QuantStudio^®^ 6 Flex Real-Time PCR System (Thermo Fisher Scientific, Waltham, MA, USA) [6]. Gene primers were designed based on the *T. truncatus* genome turTur1, in combination with a limited sequence of *D. delphis* (ncbi.nlm.gov/bioproject/421547, accessed on 29 December 2023) (Appendix A) [77]. *RPS9* (40S ribosomal protein S9), a very stable gene in cetaceans was used to normalize gene expression [78]. The following conditions were used to amplify 100 ng of cDNA: 120 s at 50 °C, 600 s at 95 °C, then 40 cycles: 15 s at 95 °C and 60 s at 60 °C. Data files were imported into ExpressionSuite Software v1.0.4 (Applied Biosystems, Foster City, CA, USA) to analyze relative expression across all plates using the comparative *Ct* method [79]. After data normalization, fold changes for *Pc^V1^* and *Fc* were calculated using the cerebellum as a calibrator. Brain-region-specific gene expression values for the harbor porpoise were compared to analogous brain regions in humans (parietal cortex, prefrontal cortex and cerebellum) using the BioGPS database (Scripps Research Institute, San Diego, CA, USA). The following NCBI Gene IDs were evaluated in BioGPS: *APP* (351), *PSEN1* (5663), *PSEN2* (5664), *MAPT* (4137), *GRN* (2896), *TARDBP* (23435) and *C9orf72* (203228); http://biogps.org/; accessed on 27 December 2023 [80].

### 5.4. Immunohistochemistry and Digital Pathology

Formalin-fixed paraffin-embedded brain tissue sections (7 μm) were prepared using a Leica microtome (Leica Biosystems, Deer Park, IL, USA) and mounted to positive charge slides as previously described [6]. Tissue sections were dehydrated, deparaffinized and placed into 3% hydrogen peroxide for 10 min to inhibit endogenous peroxidase activity. Antigen retrieval was performed by immersing brain tissue sections into heated 10 mM citric acid (pH 6.0) for 30 min followed by cooling in deionized water. For Tau AT8 immunostaining, slides were submerged into formic acid (Sigma-Aldrich, St. Louis, MO, USA) for 5 min to perform antigen retrieval. Following antigen retrieval, 5% goat serum (Vector Laboratories, Newark, CA, USA) was applied to tissues sections for 20 min to block nonspecific antibody binding. The following primary antibodies were incubated on tissue sections overnight at 4 °C: β-Amyloid 1–16 (6E10; 1.25 μg/mL; BioLegend, San Diego, CA, USA), Tau AT8 (pSer202/Thr205; 0.4 μg/mL; Thermo Fisher Chemicals, Waltham, MA, USA), TDP-43 (pSer409/410; 1.0 μg/mL; Cosmo Bio, Carlsbad, CA, USA), TARDBP (A01; 1.0 μg/mL; Abnova, Walnut, CA, USA), P62/SQSTM1 (1.0 μg/mL; Abnova, Walnut, CA, USA) and MAP2 (0.5 μg/mL; Sigma, St. Louis, MO, USA) in phosphate-buffered saline (PBS). The next day, goat-anti-rabbit IgG or horse-anti-mouse IgG biotinylated secondary antibodies were applied to brain sections (15 μg/mL; Vector Laboratories, Newark, CA, USA) for 60 min before adding the avidin–biotin complex for an additional 60 min (1:200, ABC; Vector Laboratories, Newark, CA, USA). To visualize the reaction, 3,3′-diaminobenzidine or DAB (Millipore Sigma, Burlington, VT, USA) was added to brain sections for 10 min. Postmortem brain tissue sections from AD patients were used as positive pathology controls. Archived formalin-fixed autopsy cerebral cortex samples from brain donors with intermediate to high AD (W/F/84; PMI: 16.3 h), ALS (W/M/63; PMI: 33.3 h) and FTD (W/F/68; PMI: 29.6 h) were used for comparative TDP-43 neuropathology. The donated human tissues were acquired from the University of Miami Brain Endowment Bank, an NIH NeuroBioBank under the IRB 19920348. A common dolphin (ID: IFAW 12-228 Dd) with a low BMAA tissue concentration was used to demonstrate non-pathogenic TDP-43 nuclear protein expression [7]. Immunostaining in the absence of the primary antibody was used as a negative control. Sevier Münger silver staining of harbor porpoise tissue sections was performed by AML Laboratories using an American MasterTech special kit (American MasterTech, Lodi, CA, USA) [81]. A 40× digital scan of each slide was generated using an EasyScan Pro 6 (Motic, Schertz, TX, USA) and exported to ObjectiveView^TM^ (Objective Pathology, Halton Hills, ON, Canada) and FIJI ImageJ VER2.00-rc-69/1.52p (National Institute of Health, USA) for analysis. A 2 × 5 grid totaling 1 mm^2^ was applied to the cortical layers II-VI, the CA2 field of the Hipp and the Purkinje cell layer of the CE. A semiquantitative scale was used to define pathology: (−) negative; (−/+) rare or sparse; (+) positive.

### 5.5. Statistical Analyses

Two-way ANOVA with Šídák’s multiple comparison test and Pearson r tests with a significance level of alpha = 0.05 were performed using Prism Version 9.5.1 (528) (Graph Pad, Boston, MA, USA). The normality of data was determined using the Shapiro–Wilk test. Data are presented as the mean ± standard error. 

## Figures and Tables

**Figure 1 toxins-16-00042-f001:**
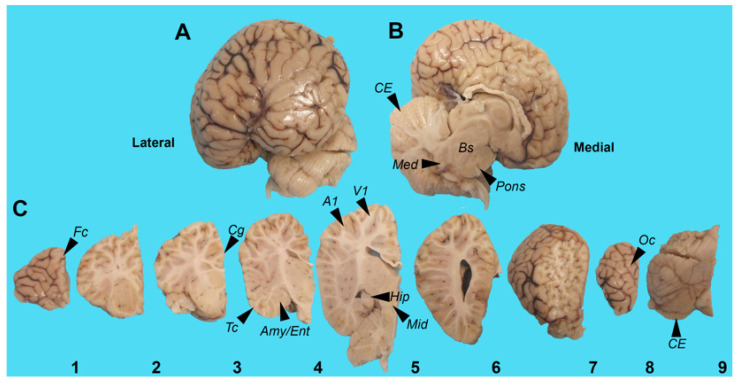
Gross examination of the harbor porpoise brain. Gross examination was performed on the left hemisphere of a beach harbor porpoise (*Phocoena phocoena*; IFAW12-206 Pp). (**A**) Lateral view of intact cerebrum and cerebellum. (**B**) Medial view of intact cerebrum, cerebellum (CE) and brainstem (Bs) containing the Pons and the medulla oblongata (Med). (**C**) Coronal sections of cerebrum (1 to 8 left to right) and detached cerebellum (9). Tissue samples were taken for histopathological analysis: 1: frontal cortex (Fc); 3: cingulate cortex (Cg); 4: amygdala (Amy), entorhinal cortex (Ent) and Temporal cortex (Tc); 5: hippocampus (Hipp), midbrain (Mid) and two gyri of the parietal cortex (Pc): auditory (A1) and visual cortex (V1). 8: occipital cortex (Oc) 9: CE.

**Figure 2 toxins-16-00042-f002:**
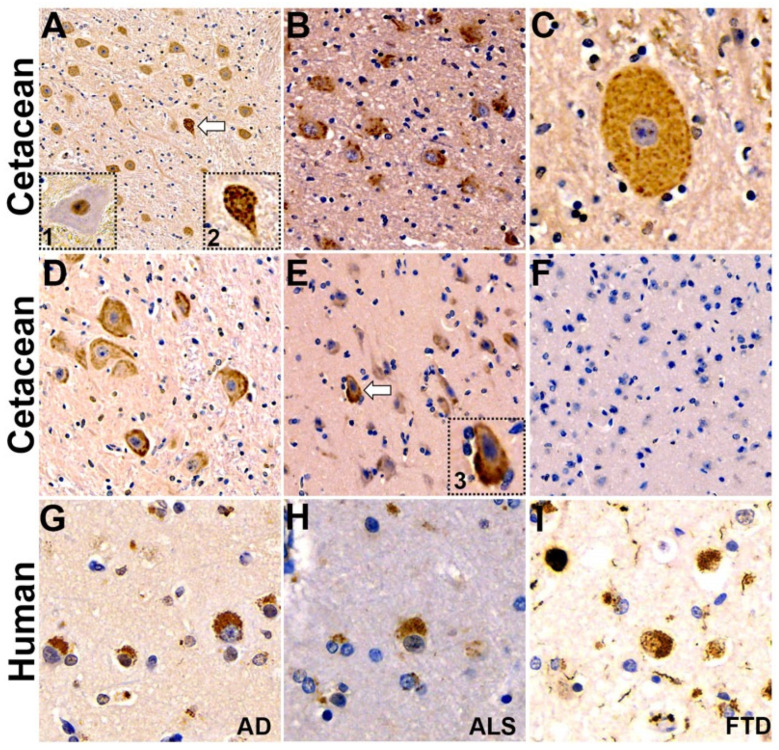
Comparative TDP-43 proteinopathy in the brain of a harbor porpoise exposed to the cyanobacterial toxin BMAA. Pathological diffused type cytoplasmic inclusions (CI) of TDP-43 were observed throughout the harbor porpoise brain. (**A**) medulla oblongata (Med); arrow indicates a neuron with severe density of TDP-43 CI. Inserts depict higher magnification images of neurons with normal (dotted box 1; from dolphin brain; low BMAA exposure) and pathological (dotted box 2; from harbor porpoise brain) expression of TDP-43. (**B**,**C**) Low- and high-magnification images of neurons in the pons region of brainstem. Images of neurons in the midbrain (Mid) (**D**) and frontal cortex (Fc) neurons in layers II–VI (**E**) that have TDP-43 CI. A higher-magnification image of a neuron in the Fc with is severe density of TDP-43 CI (arrow) depicted inset (dotted box 3). (**F**) TDP-43 CI were not observed in the occipital cortex (Oc). (**G**–**H**) TDP-43 CI observed in the Crtx of individuals with end stage Alzheimer’s disease (AD), amyotrophic lateral sclerosis (ALS) and frontal temporal dementia (FTD). Magnification: 10× (**A**); 20× (**B**–**F**); 60× (**C**); 40× (**D**–**I**); Inserts 1–3 (40×).

**Figure 3 toxins-16-00042-f003:**
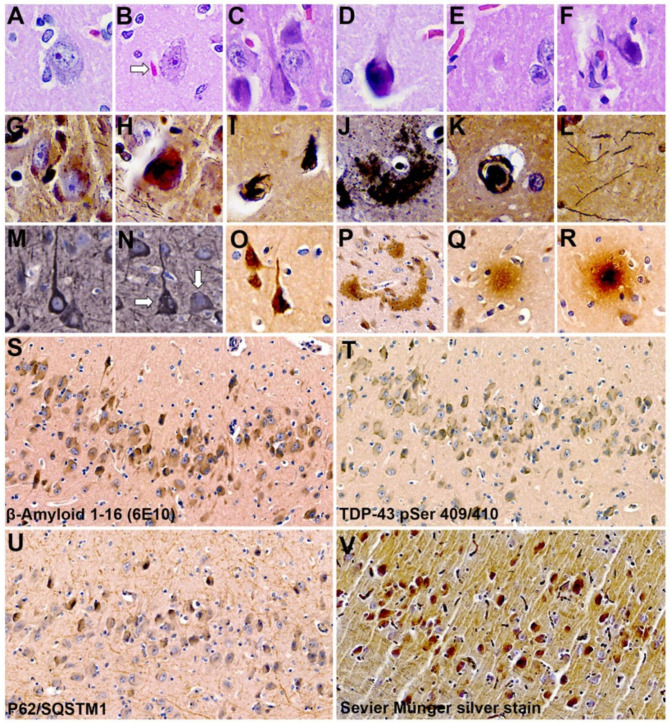
Alzheimer’s disease-like changes in a harbor porpoise exposed to the cyanobacterial toxin BMAA. H&E stains of the hippocampal pyramidal layers (**A**–**E**). Normal hippocampal pyramidal neuron (**A**), pyramidal neuron exhibiting granulovacuolar degeneration adjacent to a Hirano body (arrow) (**B**). Pyramidal neurons with the classic “flame-shaped” intracellular tangle (**C**). Pyramidal neuron with a globose tangle (**D**). Eosinophilic and plaques (**E**,**F**). Modified silver stains highlight argyrophilic intracellular inclusions (**G**,**H**), ghost tangles (**I**), diffuse plaques (**J**), a dense-core plaque (**K**) and neuropil threads (**L**). MAP2 immunostains highlight normal (**M**) and degenerative pyramidal neurons (**N**, arrows) in the frontal cortex. Aβ immunostains demonstrate an Aβ^+^ neuron (**O**), diffuse plaques (**P**) and dense-core plaques (**Q**,**R**) in the entorhinal cortex and hippocampus. Motic digital scans of CA2 hippocampal fields immunoprobed with Aβ 6E10 (**S**), pathogenic TDP-43 pSer 409/410 (**T**), P62/sequestosome-1 (P62/SQSTM1) (**U**) and Sevier Münger silver stain (**V**). Magnification: 40× (**A**–**R**); 20× (**S**–**V**).

**Table 1 toxins-16-00042-t001:** BMAA and BMAA structural isomer detection in the brain of a beached harbor porpoise.

Brain Regions	BMAA (μg/g)	AEG (μg/g)	2,4-DAB (μg/g)
Pc^A1^ (*n* = 4)	93.6 ± 9.1 *^ns^*	36.5 ± 7.7 *^p^* ^= 0.001^	282.6 ± 21.2 *^ns^*
Pc^V1^ (*n* = 5)	93.1 ± 7.5	*ND*	245.0 ± 15.4
Mean + S.E.	93.3 ± 5.4	16.2 ± 7.2	261.7 ± 13.6

A1: auditory cortex; *ND*: not detected; *ns*: no statistical significance; Pc: parietal cortex; V1: visual cortex.

**Table 3 toxins-16-00042-t003:** Histological results for brain tissue examined from a beached harbor porpoise exposed to BMAA.

Brain Regions	TDP-43 pSer 409/410	β-Amyloid 1–16 (clone 6E10)	P62/SQSTM1	SM: Argyrophilic Neurons and Plaques
Amy	+	+	+	+
CE	(−)/+	+	++	+
Cg	+	+	(−)	+
Ent	+	+	+	+
Fc	+	+	−/+	+
Hipp	+	+	+	+
Med	+	+	(−)	+
Mid	+	+	(−)	+
Oc	(−)	+	(−)	+
Pc^A1^	+	+	(−)	+
Pons	+	+	(−)	+

(−): negative; −/+: sparse; +: positive; SM: Sevier Münger; P62/SQSTM1: sequestosome 1.

## Data Availability

All data are present in the article.

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
