# Peer review of "TDP-43 and Alzheimer’s Disease Pathology in the Brain of a Harbor Porpoise Exposed to the Cyanobacterial Toxin BMAA"

_toxins, 2024, doi:10.3390/toxins16010042_

Round 1
Reviewer 1 Report
Comments and Suggestions for Authors
The manuscript is very well written and contains interesting and publishable data.
The cyanobacterial neurotoxin, BMAA and its isomers, has become ubiquitous and detected throughout the world. Unfortunately, little data on health risks for humans and animals have been reported in the scientific literature; in particular, the development of neurodegenerative diseases such as cerebral neuropathological changes like Alzheimer's disease, which could be linked to chronic exposure to BMAA. Although the data presented in this manuscript, relating to the analysis and detection of BMAA and its derivatives in the brain as well as the analysis of different neurotoxicity biomarkers, were reported in a single individual (a subadult female harbor porpoise found washed up in Massachusetts); they are interesting and confirm numerous other data published in humans and animals; and supporting Alzheimer's disease-type brain neuropathological changes associated with chronic exposure to BMAA.In addition, the experimental approach is well detailed and the results have been well discussed in relation to adequate and recent references. Therefore, I consider that this manuscript contains data of interest to the scientific community of cyanobacteriologists and publishable in this form
Author Response
Dear Reviewer,
We are very grateful for your review of our manuscript toxins-2772882 entitled: “TDP-43 and Alzheimer’s disease pathology in the brain of a harbor porpoise exposed to the cyanobacterial toxin BMAA”. Thank you for your comments, suggestions and concerns. We are grateful for your consideration of our manuscript.
With sincere gratitude,
The Authors
Reviewer 2 Report
Comments and Suggestions for Authors
The biggest limitation of this study is that it is a single-porpoise case study and it is hard to interpret what is seen in the absence of any control group. The authors might consider referring to results from the Dolphin Brain Atlas maintained by Michigan State University (the peer reviewer is uninvolved in that resource) as a way of providing some additional context for how the studied dolphin may deviate from other healthy dolphins.
The introduction and discussion cites studies consistent with the motivating hypothesis, but the BMAA literature is notoriously heterogeneous due to differences in laboratory methods and other methodological differences between research groups. If there are relevant contradictory studies regarding the premise this should be acknowledged in the introduction.
The gene expression results are a bit difficult to interpret when coming from a single dolphin, and it is unclear from the presentation what the P value refers to (what is the null hypothesis test here). It looks from Methods text like the gene expression measure is the ratio of gene expression in the target gene in target tissue vs. cerebellum compared to the gene expression ratio for 40S ribosome expression in the target tissue vs. cerebellum. Although the authors' claim is technically correct "These data suggest that the harbor porpoise expressed genes involved in the development of AD neuropathology and TDP-43 proteinopathies", it's really hard to interpret this as indicative of any pathology within the animal tending toward those disease phenotypes: a lot of genes are differentially expressed across healthy tissues, a lot of genes have pleiotropic effects nonspecific to particular diseases, and it is likely that genetic variants in these genes would cause pathology because the genes are expressed normally in healthy animals/persons as well as (a different level or form in) diseased animals/persons. This single-porpoise study has no control group for whether the observed gene expression is different from what would be expected between cerebellum and other tissues in a healthy wildtype animal. The authors are encouraged to check out the free Novartis BioGPS tool to see where gene expression differs between tissues in humans for these genes, to underscore that expression differences between tissues are normal and not necessarily reflective of any pathology. APP for example normally shows higher expression in adrenal cortex, colon, pons, cingulate cortex, and prostate than in the caudatanucleus. I did a quick query of Medlineplus.gov to see what these genes are. The normal function of APP is unclear but variants in this have been associated with Alzheimers Disease (<10% of early-onset cases), and also with hereditary cerebral amyloid angiopathy which is also a cognitive decline disorder. PSEN1 and PSEN2 encode a complex called gamma secretase that cleaves proteins. Variants in PSEN1 are not only associated with Alzheimer's Disease but also associated with a skin disorder and familial dialated cardiomyopathy (the latter is also associated with variants in PSEN2). GRN is expressed throughout the body and variants in it have been associated with CLN11 disease (seizures, vision loss, balance issues, etc.) and GRN-related frontotemporal lobar degeneration. MAPT variants have been associated with lung disease (idiopathic pulmonary fibrosis) as well as several brain diseases. TARDBP variants have been associated with ALS. C9orf72 variants have been associated with ALS. Overall, I'm not sure what we learn from seeing that the target tissue expresses these genes more than in the cerebellum in one dead porpoise. If there were 14 statistical tests being performed (frontal cortex vs. cerebellum and visual cortex vs. cerebellum for 7 candidate genes) the Bonferroni significance threshold would be 0.003571429 (=0.05/14), most of the between-tissue differences reported are Bonferroni-significant but some are not.
The pathology and imaging results are more compelling but again it is unclear to the reviewer what healthy control tissue should look like. The staining and immunoreactivity assays are somewhat compelling.
Lastly, the claim opening the Discussion that cyanobacterial HABs are increasing worldwide (attributed to climate change, nutrient pollution, and industrial waste) is an oft-repeated oversimplification. See "Perceived global increase in algal blooms is attributed to intensified monitoring and emerging bloom impacts" https://www.nature.com/articles/s43247-021-00178-8 .
Author Response
Dear Reviewer,
We are very grateful for your review of our manuscript toxins-2772882 entitled: “TDP-43 and Alzheimer’s disease pathology in the brain of a harbor porpoise exposed to the cyanobacterial toxin BMAA”. Thank you for your comments, suggestions and concerns. Below is a point-by-point response for your review and consideration. We are grateful for your consideration of our manuscript.
With sincere gratitude,
The Authors
Point-By-Point Response
1.) The biggest limitation of this study is that it is a single-porpoise case study and it is hard to interpret what is seen in the absence of any control group. The authors might consider referring to results from the Dolphin Brain Atlas maintained by Michigan State University (the peer reviewer is uninvolved in that resource) as a way of providing some additional context for how the studied dolphin may deviate from other healthy dolphins.
Author’s Response: We have added the University of Michigan Brain Biodiversity Bank dolphin brain atlas for comparative reference to the manuscript. Page 4 Lines paragraph 1 and Page 9 Lines paragraph 1.
2.) The introduction and discussion cite studies consistent with the motivating hypothesis, but the BMAA literature is notoriously heterogeneous due to differences in laboratory methods and other methodological differences between research groups. If there are relevant contradictory studies regarding the premised his should be acknowledged in the introduction.
Author’s Response: BMAA detection in seafoods and marine mammals have been well documented. To the author’s knowledge, there are no contradictory studies for BMAA detection in brain tissue of marine apex predators. However, there are contradicting studies regarding the BMAA detection in patients with Alzheimer’s disease. We have added a statement and two supporting references. Page 2 paragraph 2.
- Meneely, J.P.; Chevallier, O.P.; Graham, S.; Greer, B.; Green, B.D.; Elliott, C.T. be-ta-methylamino-L-alanine (BMAA) is not found in the brains of patients with confirmed Alzheimer's disease. Sci Rep 2016, 6, 36363, doi:10.1038/srep36363.
- Montine, T.J.; Li, K.; Perl, D.P.; Galasko, D. Lack of beta-methylamino-l-alanine in brain from controls, AD, or Chamorros with PDC. Neurology 2005, 65, 768-769, doi:10.1212/01.wnl.0000174523.62022.52.
3.) The gene expression results are a bit difficult to interpret when coming from a single dolphin, and it is unclear from the presentation what the P value refers to (what is the null hypothesis test here). It looks from Methods text like the gene expression measure is the ratio of gene expression in the target gene in target tissue vs. cerebellum compared to the gene expression ratio for 40S ribosome expression in the target tissue vs. cerebellum. Although the authors' claim is technically correct "These data suggest that the harbor porpoise expressed genes involved in the development of AD neuropathology and TDP-43 proteinopathies",it's really hard to interpret this as indicative of any pathology within the animal tending toward those disease phenotypes: a lot of genes are differentially expressed across healthy tissues, a lot of genes have pleiotropic effects nonspecific to particular diseases, and it is likely that genetic variants in these genes would cause pathology because the genes are expressed normally in healthy animals/persons as well as (a different level or form in) diseased animals/persons. This single-porpoise study has no control group for whether the observed gene expression is different from what would be expected between cerebellum and other tissues in a healthy wildtype animal. The authors are encouraged to check out the free Novartis BioGPS tool to see where gene expression differs between tissues in humans for these genes, to underscore that expression differences between tissues are normal and not necessarily reflective of any pathology. APP for example normally shows higher expression in adrenal cortex, colon, pons, cingulate cortex, and prostate than in the caudate nucleus. I did a quick query of Medlineplus.gov to see what these genes are. The normal function of APP is unclear but variants in this have been associated with Alzheimer’s Disease (<10% of early-onset cases), and also with hereditary cerebral amyloid angiopathy which is also a cognitive decline disorder. PSEN1 and PSEN2encode a complex called gamma secretase that cleaves proteins. Variants inPSEN1 are not only associated with Alzheimer's Disease but also associated with a skin disorder and familial dilated cardiomyopathy (the latter is also associated with variants in PSEN2). GRN is expressed throughout the body and variants in it have been associated with CLN11 disease (seizures, vision loss, balance issues, etc.) and GRN-related frontotemporal lobar degeneration. MAPT variants have been associated with lung disease (idiopathic pulmonary fibrosis) as well as several brain diseases. TARDBP variants have been associated with ALS. C9orf72 variants have been associated with ALS. Overall, I'm not sure what we learn from seeing that the target tissue expresses these genes more than in the cerebellum in one dead porpoise.
Author’s Response: The null hypothesis tests that dementia related genes in humans with disease (AD, ALS, FTD, etc.) are expressed in the porpoise brain. This data to our knowledge has not been reported. Expression from two cortical regions are normalized to the cerebellum, a region usually unaffected in human neurodegenerative disease, to show a regions specific pattern of expression. We did not evaluate other forms of gene variants or mutations, etc. The goal of this data is to show that these genes are expressed the same in the porpoise as they are in humans in order to stimulate further research into how cyanotoxins impact these genes. We have updated our text to make this cleared to the reader. In addition, we have added comparative human expression values of cortex normalized to cerebellum from BioGPS to show that genes of interest are expressed in a similar manner in the porpoise. Table 2 has also been updated to delineate the type of neuropathological lesion associated with the genes of interest. Page 3 paragraph 2.
4.) If there were 14 statistical tests being performed (frontal cortex vs. cerebellum and visual cortex vs. cerebellum for 7 candidate genes) the Bonferroni significance threshold would be 0.003571429 (=0.05/14), most of the between-tissue differences reported are Bonferroni-significant but some are not.
Author’s Response: The statistical analysis for TABLE 2 has been updated using a two-way ANOVA comparing brain region (Pcv1 & Fc) vs. gene expression (7 genes) using Šídák's multiple comparisons test. Page 3 paragraph 2.
5.) The pathology and imaging results are more compelling but again it is unclear to the reviewer what healthy control tissue should look like. The staining and immunoreactivity assays are somewhat compelling.
Author’s Response: Although, we could not find a control porpoise that is BMAA-free, we used archived brain tissue sections from a closely related cetacean species Delphinus delphinus with very low BMAA exposure ~ 0-20 ug/g to display how non-pathological TDP-43 immunostaining looks in a healthy neurons. In addition, we also added comparative pathology from individuals with neuropathological confirmed AD, ALS, and FTD. Page 5. figure 2
6.) Lastly, the claim opening the Discussion that cyanobacterial HABs are increasing worldwide (attributed to climate change, nutrient pollution, and industrial waste) is an oft-repeated oversimplification. See "Perceived global increase in algal blooms is attributed to intensified monitoring and emerging bloom impacts"https://www.nature.com/articles/s43247-021-00178-8 .
Author’s Response: Thank you for sharing this wonderful reference. We removed the broad global statement from the abstract and discussion and added a sentence about regional variation and referenced this paper. Page 7 paragraph 1.
Reviewer 3 Report
Comments and Suggestions for Authors
In this manuscript, the Authors presented the effect of cyanobacterial toxin b-N-methylamino-L-alanine (BMAA) on TDP-43 and Alzheimers disease pathology in the brain of harbor porpoise. The case history, analysis and representation of gene expression and immunohistochemistry are presented with proper methodology. Interestingly, the discussion part is represented with more relevant references. The results reflect the title and conclusion of present manuscript. So, I recommend this manuscript for acceptance in the present form.
Comments on the Quality of English LanguageEnglish language can be improved in terms of continuation of sentences.
Author Response
Dear Reviewer,
We are very grateful for your review of our manuscript toxins-2772882 entitled: “TDP-43 and Alzheimer’s disease pathology in the brain of a harbor porpoise exposed to the cyanobacterial toxin BMAA”. Thank you for your comments, suggestions and concerns. We are grateful for your consideration of our manuscript.
We have updated some the critical text as suggested in the revised manuscript.
With sincere gratitude,
The Authors